# Molecular-Scale Liquid Density Fluctuations and Cavity Thermodynamics

**DOI:** 10.3390/e26080620

**Published:** 2024-07-24

**Authors:** Attila Tortorella, Giuseppe Graziano

**Affiliations:** 1Scuola Superiore Meridionale, Via Mezzocannone, 4, 80138 Naples, Italy; attila.tortorella@unina.it; 2Department of Chemical Sciences, University of Naples Federico II, Via Cintia, 4, 80126 Naples, Italy; 3Department of Science and Technology, University of Sannio, Via Francesco de Sanctis, snc, 82100 Benevento, Italy

**Keywords:** density fluctuations, maximum entropy principle, Gaussian distribution, cavity distribution, solvent-excluded volume effect

## Abstract

Equilibrium density fluctuations at the molecular level produce cavities in a liquid and can be analyzed to shed light on the statistics of the number of molecules occupying observation volumes of increasing radius. An information theory approach led to the conclusion that these probabilities should follow a Gaussian distribution. Computer simulations confirmed this prediction across various liquid models if the size of the observation volume is not large. The reversible work required to create a cavity and the chance of finding no molecules in a fixed observation volume are directly correlated. The Gaussian formula for the latter probability is scrutinized to derive the changes in enthalpy and entropy, which arise from the cavity creation. The reversible work of cavity creation has a purely entropic origin as a consequence of the solvent-excluded volume effect produced by the inaccessibility of a region of the configurational space. The consequent structural reorganization leads to a perfect compensation of enthalpy and entropy changes. Such results are coherent with those obtained from Lee in his direct statistical mechanical study.

## 1. Introduction

A theoretical analysis of solvation (i.e., the transfer of a solute molecule from a fixed position in the gas phase to a fixed position in the liquid phase, according to the so-called Ben–Naim standard [1]) indicates the need to take into account the cavity creation process in the solution [2,3,4,5,6,7]. The need descends from the recognition that, since a liquid is a condensed state of the matter, a suitable space (i.e., a cavity) must be created, at a fixed position, to host the solute molecule. This need is well understood by theoreticians but not so well by experimentalists. The latter claim that any liquid possesses a lot of void space, around 50% of the total volume, and so cavity creation should be unnecessary. However, the void volume in a liquid is divided into many tiny pieces whose dimensions are not able to host real solute molecules [8,9,10,11,12] (for instance, the average dimensions of such small pieces depend on the diameter of liquid molecules, which can be understood by thinking of the voids left in a box filled by tennis balls or by ping-pong balls). The process of cavity creation can be examined through theoretical approaches or computer simulations using appropriate models for different liquids. The reversible work associated with cavity creation is a positive and large quantity for every liquid system. [2,3,4,5,6,7]. Moreover, it is accepted that the cavity creation reversible work possesses its largest value in water compared to other liquid systems [13,14,15], which is also the reason for the low solubility of nonpolar molecules in water [2,3,4,5,6,7].

Cavity creation was analyzed by means of statistical mechanics 40 years ago by Lee [16], and there were two findings. (a) The reversible work of cavity creation has an entropic origin, since cavity creation reduces the number of configurations available to liquid molecules (i.e., only the liquid configurations in which the described cavity is present can be chosen). Thus, the statistical ensemble size is reduced, leading to an entropy decrease in all liquids, which can be described as a solvent-excluded volume effect. (b) There is a cavity enthalpy change coming from a structural reorganization of liquid molecules (cavity creation is a perturbation that pushes the liquid molecules to assume positions which render possible the cavity existence). This structural reorganization differs from the solvent-excluded volume effect and produces also an entropy change that exactly compensates the cavity enthalpy change [16,17]. The creation of cavities is only driven by entropy, which is a fundamental characteristic that applies to all cavities formed by molecular-scale equilibrium density fluctuations.

A fundamental theorem in statistical mechanics relates the reversible work of creating a cavity to the logarithm of the probability, *P*(0; v), of finding no liquid molecules within the volume of the desired cavity [18]. In order to exploit this connection, *P*(0; v) must be known. Pratt and colleagues [19,20,21] devised an elegant information theory approach to arrive at *P*(0; v) by determining the probabilities of finding the centers of *n* molecules *P*(*n*; v) inside a randomly positioned volume v, which corresponds to the solvent-excluded volume of the specified cavity (i.e., a spherical cavity has a van der Waals radius r_c_ and a solvent-excluded radius R_c_ = r_c_ + r_s_, where r_s_ is the radius which describes the solvent as spheres; in other words, r_c_ is the radius of the spherical volume where no part of the solvent molecules can be found, while R_c_ is the radius of a sphere where no center of the solvent molecules can be found [17]). Using a flat default model in an approach based on the maximum entropy principle [22] resulted in a discrete Gaussian distribution for the *P*(*n*; v) probabilities [19]. This theoretical result is in line with the fluctuation theory in statistical mechanics where a Gaussian approximation holds for the fluctuation in the number of particles in the grand-canonical ensemble [23]. Moreover, it has been supported by computer simulations of liquid models (i.e., hard sphere fluids [24], Lennard–Jones liquids [25], n-hexane [26], dimethyl sulfoxide [26], and several models of water [19,27,28,29]). The *P*(*n*; v) probabilities are well described by Gaussian distributions for not so large observation volumes (i.e., when the ratio of observation volume radius to liquid molecule radius is smaller than two, the Gaussian distribution holds regardless of the nature of the energetic interactions among liquid molecules). For instance, it has been shown that the so-called monoatomic water model [30] is characterized by equilibrium density fluctuations that follow a Gaussian distribution up to a cavity radius R_c_ ≈ 4 Å [31]. It should be clear that there is no compelling reason to expect that the *P*(*n*; v) probabilities should obey a Gaussian distribution. Indeed, by increasing the radius of the observation volume in water models, there are large deviations from Gaussian values in the low-number tail of the distribution [27,28,29,31,32,33]. Moreover, a different distribution, called a binomial cell model, has been proposed to analyze molecular-scale density fluctuations in water models [34].

Notwithstanding the studies published on this matter, the basic relationships between the probability distribution of number density fluctuations and cavity thermodynamics have not yet spelled out in detail except for the analysis by one of us [35]. In this study, we want to demonstrate that the entropic basis of the reversible work of cavity creation directly emerges by the *P*(0; v) formula provided by the Gaussian distribution.

## 2. Theoretical Foundation

A liquid possesses a huge ensemble of molecular configurations, and a statistical mechanical description is unavoidable. Assuming that *X* is a multidimensional vector accounting for the coordinates of every spherical molecule which is present in the liquid, the probability density function associated with a specific liquid configuration, in the NPT ensemble, is [17]
ρ(***X***) = exp[−H(***X***)/*k*T]/∫ exp[−H(***X***)/*k*T]d***X***(1)
where H(***X***) = E(***X***) + P·V(***X***) is the enthalpy of the configuration ***X***, E(***X***) represents the total interaction energy of liquid molecules in the configuration ***X***, V(***X***) is the volume of the configuration ***X***, P is the pressure of the liquid, *k* is the Boltzmann constant, and the denominator is the isobaric–isothermal configurational partition function. The probability of cavity occurrence in the liquid is the chance of finding no molecular centers in the solvent-excluded volume, v, of the cavity, or the chance that the centers of all the N spherical molecules of the liquid are located outside the solvent-excluded volume, v, of the cavity. This probability is obtained by integrating ρ(***X***) over all the configurations having the centers of all the N spherical molecules in the volume <V> − v [2,35]. In this respect, it is important to note that (a) the position of the cavity must be fixed but can be arbitrarily located within the liquid volume since the liquid density is uniform at equilibrium; (b) although the total volume is not strictly constant in the NPT ensemble, it is reasonable to assume that the total volume of a macroscopic system will always be close to the ensemble average value ⟨V⟩. Thus, one has
*P*(0; v) = *P*(N; <V> − v) = ∫ρ(***X***) d***X***
           <V> − v(2)
where the integration domain has a clarified meaning. The calculation of *P*(0; v) corresponds to picking out only a very small fraction of the total liquid configurations, i.e., the ones having the cavity of the requested solvent-excluded volume. This selection strongly reduces the amount of molecular configurations available to the liquid and leads to a liquid entropy loss (i.e., entropy is an extensive thermodynamic function, and its magnitude depends on the size of the statistical ensemble [16,18,23]). This entropy loss holds for any liquid with no regard for the interactions between the liquid molecules.

Cavity creation in a determined position within a liquid, keeping fixed NPT, leads to an increment of the average volume of the system by an extent equal to the van der Waals volume of the cavity, v_vdW_. Nevertheless, the presence of a void sphere of solvent-excluded volume v implies that the spherical shell given by the difference (v − v_vdW_) becomes inaccessible to the centers of liquid molecules. This solvent-excluded volume effect is a constraint for every molecule of the liquid whose centers, during their continuous translations, cannot enter the cavity solvent-excluded volume if the cavity must exist (see Figure 1). The inaccessible shell may be approximated by the solvent accessible surface area of the cavity in all the solvents. It is interesting to note that a geometric entropy, linearly proportional to the superficial area of the solvent-excluded volume, emerged in a theoretical approach based on a density field theory [36].

As underscored by Tolman [18], there is an exact statistical mechanical relationship between the occurrence probability of a constrained configuration of a thermodynamic system and the reversible work to produce that constrained configuration:*P*(0; v) = exp[−W(0; v)/*k*T](3)
where W(0; v) is the reversible work (i.e., the Gibbs free energy change) to create a cavity of solvent-excluded volume equal to v, W(0; v) = ΔG_c_(v; R_c_), where R_c_ is the solvent-excluded cavity radius. It is important to underscore that at 300 K and 1 atm, Equation (3) implies that ΔG_c_ = 40.0 kJ mol^−1^ corresponds to *P*(0; v) = 1.1 10^−7^, and ΔG_c_ = 60.0 kJ mol^−1^ corresponds to *P*(0; v) = 2.0 10^−9^, regardless of the liquid. These numbers highlight how large the decrease in available liquid configurations caused by cavity creation is. Now, the assumption that equilibrium density fluctuations at a molecular level follow a Gaussian distribution can be scrutinized to shed further light on the entropy loss associated with cavity creation.

## 3. Gaussian Fluctuations

In compliance with the results of computer simulations of various liquids [19,20,21,24,25,26,27,28,29,31], the probability of finding the centers of exactly *n* molecules within an observation volume v, when the liquid number density is ρ ≡ N_Av_/v_m_ (i.e., N_Av_ is the Avogadro’s number and v_m_ is the molar volume of the liquid), and pressure and temperature are held constant, is well described by a Gaussian distribution if the observation volume is not large (see above). Therefore, one has
*P*(n; v) = (2π·σ*_n_*^2^)^−1/2^ · exp(−δ*n*^2^/2σ*_n_*^2^)(4)
where δ*n* = *n* − <*n*>, <*n*> is the average number of molecular centers in the volume v, <*n*> = ρ·v, and σ*_n_*^2^ = <δ*n*^2^> = <*n*^2^> − <*n*>^2^ is the variance of the Gaussian distribution, i.e., the mean square fluctuation in the number of molecular centers inside the volume v. It is important to recognize that the first two moments of the Gaussian distribution are related to the number density and the radial distribution function of the liquid, respectively, which are quantities that are experimentally measurable [19,20,21]. Since we are looking for the probability of finding a cavity in the liquid, we need the probability *P*(0; v) of finding no molecular centers in the volume v:*P*(0; v) = (2π·σ*_n_*^2^)^−1/2^ · exp(−<*n*>^2^/2σ*_n_*^2^) = (2π·σ*_n_*^2^)^−1/2^ · exp(−ρ^2^v^2^/2σ*_n_*^2^)(5)The probability of cavity occurrence is related to equilibrium density fluctuations on a molecular level, underscoring that the creation of a cavity is a special process, which only depends on the properties of the pure liquid. Introducing Equation (5) into Equation (3), one obtains the following:ΔG_c_(v; R_c_) = (*k*T/2)·ln(2π·σ_n_^2^) + (*k*T· ρ^2^v^2^/2σ_n_^2^)(6)The σ_n_^2^ value depends on the v size and can solely be determined by means of computer simulations on a molecularly detailed model of the liquid of interest [19,20,21]. According to the values reported by Sulimov and co-workers in the case of the TIP4P water model [34], for a cavity whose solvent-excluded volume is suitable to host methane, R_c_ = 3.3 Å, <*n*> = 5.11, σ_n_^2^ = 1.39 [34], and using Equation (6), ΔG_c_ = 26.1 kJ mol^−1^ at 300 K; for a cavity whose solvent-excluded volume is suitable to host neopentane, R_c_ = 4.4 Å, <*n*> = 11.77, σ_n_^2^ = 2.62 [34], and using Equation (6), ΔG_c_ = 69.4 kJ mol^−1^ at 300 K [34]. These ΔG_c_ values are in line with those calculated by direct computer simulations, as it can readily be controlled upon looking at Table 3 in [5]. Moreover, it is easy to verify that on increasing the cavity solvent-excluded volume, the logarithmic term in Equation (6) becomes smaller and smaller in comparison to the other one. So it is possible to state that ΔG_c_ is inversely proportional to the variance of the Gaussian distribution, ΔG_c_ ∝ 1/σ_n_^2^. We do not want to perform calculations with Equation (6) but rather to deepen its statistical and thermodynamic features and consequences.

The joint probability of having both a cavity of solvent-excluded volume v_1_ and a cavity of solvent-excluded volume v_2_ can be given by a Gaussian distribution of solvent-excluded volume v_1_ + v_2_. The latter Gaussian distribution, however, is not the product of the two Gaussian functions describing the probability of zero occupancy in v_1_ and zero occupancy in v_2_:*P*(0; v_1_ + v_2_) ≠ *P*(0; v_1_)·*P*(0; v_2_)(7)The two events do not appear to be independent of each other because the zero occupancy of v_1_ depends on the occupancy number of v_2_, since the number of liquid molecules is fixed in the system. This feature of the Gaussian functions highlights an interesting physical point [20].

When the volume of interest corresponds to the molar volume of the liquid, the variance of the Gaussian distribution is proportional to the isothermal compressibility, σ_n_^2^ ∝ β_T_ [23], and this implies that ΔG_c_ ∝ 1/β_T_. The reversible work of cavity creation, measuring the entropy loss due to the solvent-excluded volume effect, is inversely proportional to the isothermal compressibility of the liquid. This relationship was originally obtained by Pratt and colleagues [19,20,21]. Water has the smallest β_T_ value among all common liquids (i.e., at 25 °C, β_T_(in atm^−1^·10^5^) = 4.58 for water, 9.80 for benzene, 11.55 for c-hexane, 16.27 for n-hexane, 10.81 for carbon tetrachloride, 14.79 for methanol, and 10.26 for ethanol [17]), and, in fact, it has the largest ΔG_c_ value for a given van der Waals cavity radius among all common liquids, as demonstrated by computer simulation results [13,14,15]. The isothermal compressibility is a macroscopic thermodynamic quantity, and a closer scrutiny is necessary to single out the microscopic difference between water and the other liquids.

In compliance with statistical mechanics, β_T_ measures the fluctuations in the liquid number density in the grand canonical ensemble, β_T_ = v_m_·σ*_n_*^2^/<*n*>^2^·*k*T [23]. Water has the lowest β_T_ value among all common liquids, since the molar volume of water is the smallest among those of all common liquids: at 25 °C and 1 atm, v_m_(in cm^3^ mol^−1^) = 18.07 for water [37], 89.41 for benzene, 108.75 for c-hexane, 131.62 for n-hexane, 97.09 for carbon tetrachloride, 40.73 for methanol, and 58.68 for ethanol [38]. This fact stems from the effective size of liquid molecules. The effective size of water molecules is the smallest among those of all common liquids; at 25 °C, the effective hard sphere diameter is 2.80 Å for water [12], 5.26 Å for benzene, 5.63 Å for c-hexane, 5.92 Å for n-hexane, 5.37 Å for carbon tetrachloride [38], 3.83 Å for methanol, and 4.44 for ethanol Å [17]. The effective hard sphere diameter of water molecules is even smaller than their van der Waals diameter as a consequence of the bunching up effect due to the strength of H bonds [39].

Using fundamental relationships of equilibrium thermodynamics to Equation (6) allows the derivation of the changes of both enthalpy and entropy which arise from cavity creation:ΔH_c_ = −T^2^[*∂*(ΔG_c_/T)/*∂*T]_P_ = −(*k*T^2^/2)·{*∂*[ln(2π·σ*_n_*^2^) + (ρ^2^v^2^/σ*_n_*^2^)]/*∂*T}_P_ = = −(*k*T^2^/2σ_n_^2^)·{[1 − ((ρ^2^v^2^/σ*_n_*^2^)]·(*∂*σ*_n_*^2^/*∂*T)_P_ + 2ρv^2^·(*∂*ρ/*∂*T)_P_}
(8)
and ΔS_c_ = −(*∂*ΔG_c_/*∂*T)_P_ = −(*k*/2)·ln(2π·σ*_n_*^2^) − (*k*ρ ^2^v^2^/2σ*_n_*^2^) +  −(*k*T/2σ_n_^2^)·{[1 − ((ρ^2^v^2^/σ*_n_*^2^)]·(*∂*σ*_n_*^2^/*∂*T)_P_ + 2ρv^2^·(*∂*ρ/*∂*T)_P_}(9)In performing the derivatives, the σ_n_^2^ quantity has been considered a single variable function of temperature, and the v quantity has been considered temperature independent [35]. It is worth noting that these “approximations” affect in the same manner both ΔH_c_ and ΔS_c_ and do not alter the analysis below. Equations (8) and (9) illustrate the following points: (a) the enthalpy change associated with cavity formation is completely offset by a corresponding entropy change, resulting in no net enthalpic contribution to the reversible work of cavity creation; (b) the cavity entropy change includes an additional term, expressed as −ΔG_c_/T, which quantifies the solvent-excluded volume effect related to cavity formation in a liquid. This term represents the entropy reduction due to the decreased size of the liquid’s statistical ensemble when selecting configurations that include the desired cavity. The above sentences may appear a circular argument [40] unless Equation (8) is identified as the enthalpy change due to cavity creation in an independent manner.

In water, the σ*_n_*^2^ quantity depends little on temperature because the isothermal compressibility of water is nearly constant in the temperature range 0–100 °C [37]. Thus, the quantity (*∂*σ_n_^2^/*∂*T)_P_ should be negligible, and Equation (8) can be rearranged to
ΔH_c_ ≅ −(*k*T^2^ρv^2^/σ_n_^2^)·(*∂*ρ/*∂*T)_P_ = *k*T^2^ρ^2^v^2^α_P_/σ_n_^2^(10)
where α_P_ = −(1/ρ)·(*∂*ρ/*∂*T)_P_ is the isobaric thermal expansion coefficient of the liquid. According to Equation (10), ΔH_c_ ∝ α_P_, agreeing with the equation originally derived by Pierotti [41] within the framework of scaled particle theory [42]. In the other liquids, the quantity σ_n_^2^ depends on temperature, but the factor [1 − (ρ^2^v^2^/σ*_n_*^2^)] occurring in Equation (8) should be small; thus, Equation (10) is going to be a not-bad approximation for every liquid. By assuming that v = v_m_ and using the statistical mechanical definition of the isothermal compressibility in Equation (10), the latter becomes
ΔH_c_ = α_P_·T·v_m_/β_T_(11)Both α_P_ and β_T_ are thermodynamic response functions [23], and it is reliable to associate them with a process such as cavity creation that implies a structural reorganization of the pure liquid. In the configurations possessing the desired cavity, liquid molecules must have special spatial distributions that produce changes in both enthalpy and entropy. This structural reorganization can be described by a proper function of α_P_ and β_T_ of the pure liquid, because there is no solute molecule inserted in the liquid when the cavity is created. On the basis of Equation (11), it is correct to state that the structural reorganization (which is distinct from the solvent-excluded volume effect) associated with cavity creation is characterized by a complete enthalpy–entropy compensation. A qualitative picture of the various thermodynamic quantities associated with cavity creation in water, at room temperature and 1 atm, is shown in Figure 2.

## 4. Conclusions

Equilibrium density fluctuations at a molecular scale follow a Gaussian distribution in several liquids when the observation volumes are not large. This makes it possible to arrive at an analytical relationship for the probability of finding no molecules in a solvent-excluded volume corresponding to the desired cavity [19,20,21]. A careful analysis of this relationship leads to formulas for the changes in Gibbs free energy, enthalpy and entropy which occur upon cavity creation. These formulas demonstrate that (a) the Gibbs free energy required for the creation of a cavity is purely entropic due to the reduction in the dimensions of the statistical ensemble caused by the solvent-excluded volume effect; (b) there is a complete compensation between the enthalpy and entropy changes stemming from the rearrangement of solvent molecules around the cavity. This thermodynamic scenario matches the one determined by Lee [16] with a general statistical mechanical approach.

## Figures and Tables

**Figure 1 entropy-26-00620-f001:**
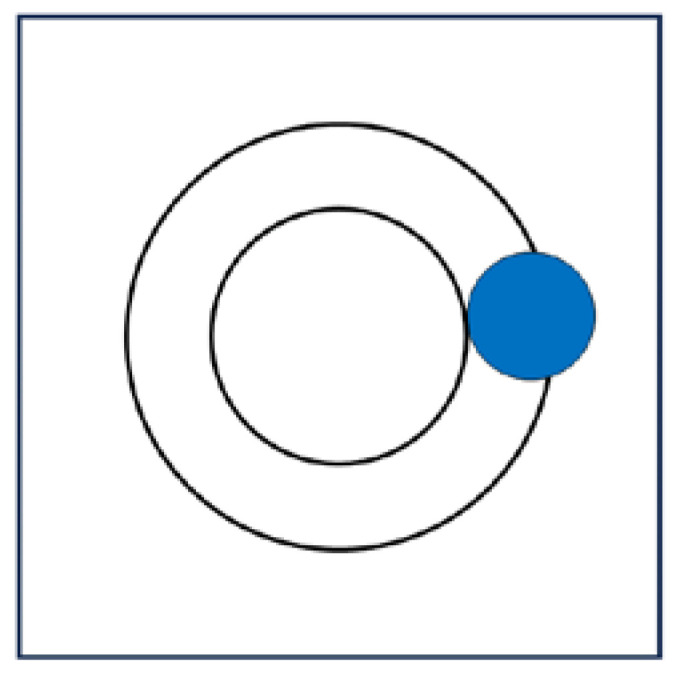
The cavity creation (i.e., the inner circle), at constant NPT, increases the volume of the system by an amount which corresponds to the van der Waals cavity volume. As a consequence, a spherical shell corresponding to the difference between the solvent-excluded volume of the cavity and its van der Waals volume (i.e., the space between the outer circle and the inner one) becomes inaccessible to the center of liquid molecules (the filled blue circle represents one liquid molecule) if the cavity is to exist. This geometric effect comes from the solvent-excluded volume associated with cavity creation.

**Figure 2 entropy-26-00620-f002:**
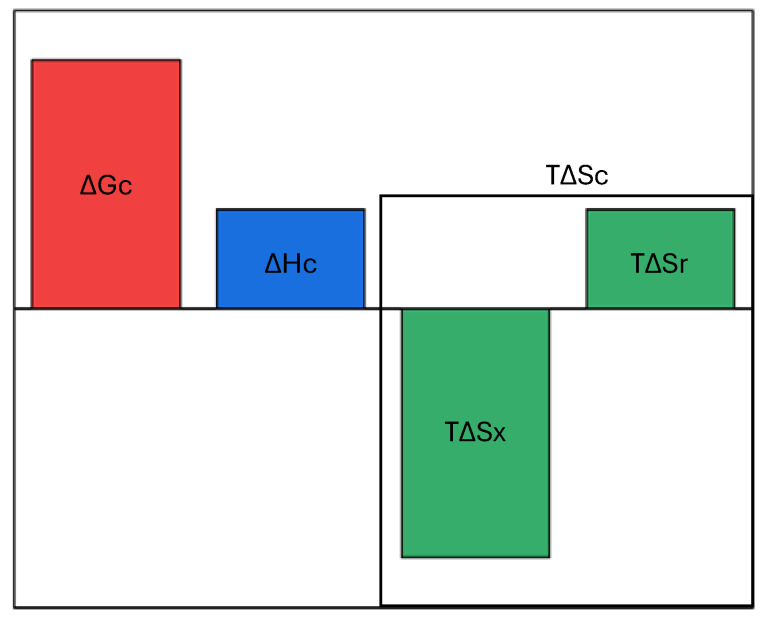
A qualitative bar plot of the various thermodynamic quantities associated with cavity creation in water at room temperature and 1 atm. The cavity entropy change has been divided in the solvent-excluded volume contribution, labeled x, and the liquid structural reorganization contribution, labeled r. The latter exactly compensates the enthalpy term.

## Data Availability

The original contributions presented in the study are included in the article, further inquiries can be directed to the corresponding author.

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
