# Peer review of "Molecular-Scale Liquid Density Fluctuations and Cavity Thermodynamics"

_entropy, 2024, doi:10.3390/e26080620_

Round 1

Reviewer 1 Report

Comments and Suggestions for Authors

The article looks at the creation of cavity in a liquid from the entropic and enthalpic points of view. It is clear and well written. The introduction describes the subject and the statistical mechanics modeling tools used to characterize the creation process. A schematic diagram would be welcome to clarify the terms enthalpy/entropy of cavity creation and net contribution to cavity creation, so as to guide the reader through these nuances.

The second section provides the theoretical basis for quantifying cavity creation. The third section applies this theory using results from the literature. It is thus possible to estimate Gibbs energy values associated with this transformation and compare them with results from other approaches, but the basic assumptions (Gaussian probability distribution) and the results for comparison come from the same works, notably reference [19]. They are therefore necessarily consistent; is it possible to have validation bases that do not come from the same source?

The expressions for the enthalpy and entropy of cavity formation (equations 7 and 8) are based on certain assumptions and so the conclusions drawn from them are also restricted to these assumptions; this needs to be clarified.

Finally, the conclusion mentions that this work is perfectly consistent with Lee's with a general statistical mechanism approach; there is no mention of the particular contribution of this work.

Reviewer 2 Report

Comments and Suggestions for Authors

please find my remarks in the attached pdf file

Round 2

Reviewer 1 Report

Comments and Suggestions for Authors

All the comments of the first review have been taken into account.